

# A New Seismicity Catalogue of the Eastern Alps Using the Temporary Swath-D Network

Laurens Jan Hofman[1], Jörn Kummerow[1], Simone Cesca[2], and the AlpArray-Swath-D Working Group [*]

[1]Earth Science Department, Section of Geophysics, Freie Universität Berlin, Germany
[2]GFZ German Research Centre for Geosciences, Potsdam, Germany
[*]For a complete list of team members, please visit the link that appears at the end of the paper

**Correspondence:** Laurens Jan Hofman (rens@geophysik.fu-berlin.de)

**Abstract.** We present a new, consistently processed seismicity catalogue for the Eastern and Southern Alps, based on the temporary dense Swath-D monitoring network. The final catalogue includes $6,053$ earthquakes for the time period 2017-2019 and has a magnitude of completeness of $M_c \sim 1.0$. The smallest detected and located events have a magnitude of $M_l = -0.5$. Aimed at the low to moderate seismicity in the study region, we generated a multi-level, mostly automatic workflow which
combines a priori information from local catalogues and waveform-based event detection, subsequent efficient GPU-based event search by template matching, P & S arrival time pick refinement and location in a regional 3-D velocity model.

The resulting seismicity distribution generally confirms the previously identified main seismically active domains, but provides increased resolution of the fault activity at depth. In particular, the high number of small events additionally detected by the template search contributes to a more dense catalogue, providing an important basis for future geological and tectonic
studies in this complex part of the Alpine orogen.

## 1 Introduction

The Alps constitute the largest and highest mountain range in Europe and are the result of the collision between the European plate and the Adriatic microplate. The influence of crustal indentation in forming the Alpine orogen was disclosed by geological-geophysical transects and passive seismic experiments across the Western, Central and Eastern Alps such as
ECORS-CROP (Nicolas et al., 1990); NFP20 (Pfiffner et al., 1990); TRANSALP (TRANSALP Working Group et al., 2002; Schmid et al., 2004); CIFALPS (Malusà et al., 2021). These and teleseismic, local earthquake & ambient-noise tomography studies have also revealed the complexity of the crustal and upper mantle structure (e.g. Piromallo and Morelli, 2003; Lippitsch et al., 2003; Bleibinhaus and Gebrande, 2006; Diehl et al., 2009; Zhao et al., 2015; Hua et al., 2017; Kissling and Schlunegger, 2018; Kästle et al., 2018; Kästle et al., 2020; Lu et al., 2020; Qorbani et al., 2020; Nouibat et al., 2021; Paffrath et al., 2021;
Jozi Najafabadi et al., 2022; Paul, 2022). Specifically in the Eastern and Southern Alps, however, key questions regarding the deeper structure such as the possible existence and location of a slab polarity remain controversial (e.g. Lippitsch et al., 2003; Kummerow et al., 2004; Handy et al., 2010; Mitterbauer et al., 2011; Kästle et al., 2020; Mroczek et al., 2023).

Another approach to illuminate the crustal structure and tectonic processes is the analysis of earthquake activity. The overall diffuse shallow seismicity in the Alps was investigated by various studies, ranging from regional to local scales (e.g. Nicolas



et al., 1998; Bethoux et al., 1998; Reinecker and Lenhardt, 1999; Chiarabba et al., 2005; Ustaszewski and Pfiffner, 2008; Anselmi et al., 2011; Bressan et al., 2012; Viganò et al., 2015; Reiter et al., 2018; Beaucé et al., 2019; Jozi Najafabadi et al., 2021; Saraò et al., 2021).

Until recently, homogeneous monitoring of the Alpine seismicity was impeded by an irregular configuration of permanent seismic stations and non-uniform procedures of the national seismological agencies. The regional-scale AlpArray initiative
(Hetényi et al., 2018) finally enabled a new coherent analysis of the earthquakes in the greater Alpine region, producing the AlpArray seismicity catalogue with a magnitude of completeness of $M_c = 2.4$ (Bagagli et al., 2022).

Embedded into AlpArray, the additional, denser Swath-D network, comprising 151 uniformly spaced stations in the Eastern and Southern Alps, was operated for approximately two years in the time period 2017-2019 (Heit et al., 2021, and Fig. 1). In this study, we exploit the new data contributed by the Swath-D network and the data from adjacent stations to build a
consistently processed, more detailed earthquake catalogue focusing on the Eastern and Southern Alps and provide the basis for a subsequent in-depth analysis of seismicity and the related fault structures.

The Swath-D region of the Alps is characterised by relatively low to moderate seismicity (e.g. Slejko et al., 1998; Reiter et al., 2018). Nonetheless, it is routinely monitored by different agencies which already provide local earthquake catalogues of high-quality (Istituto Nazionale di Oceanografia e di Geofisica Sperimentale (OGS), 2016; INGV Seismological Data Centre,
2006; Swiss Seismological Service (SED) at ETH Zurich, 1983). We therefore generated an event detection and location workflow that both integrates the already existing information and exploits the newly available data from the Swath-D monitoring network. We first build a preliminary earthquake catalogue based on the local event catalogues which we complement by an energy-based detection using the entire continuous waveform archive of the Swath-D network and waveforms of neighbouring stations in the same time period. We then search for smaller, hitherto undetected events by a template matching technique, i.e.
we scan the continuous waveform data for similarity with waveforms from the preliminary event catalogue. A number of applications have demonstrated the effectiveness of this approach to increase the number of low magnitude earthquake detections (e.g. Gibbons and Ringdal, 2006; Skoumal et al., 2015; Vuan et al., 2018; Ross et al., 2019; Beaucé et al., 2019). In order to cope with the large data volume caused by the high number of seismic stations (198, including Swath-D and adjacent stations from AlpArray and local monitoring networks), relatively long registration period (24 months) and high sampling rate (100 Hz)
of the Swath-D network, we developed an efficient, GPU-accelerated algorithm for computing the cross-correlation which we use as the measure of waveform similarity (GPU- Graphical Processing Unit). For each event cluster defined by the previous template search, we identify a reference event and manually pick P and S arrival times. We then automatically associate new picks for all cluster events by using relative arrival times determined from a combination of the cross-correlation function and the short-term-average to long-term-average ratio (STA/LTA). The extended set of P and S arrival times is then inverted for
event location and origin time in a local 3-D velocity model. Since the Eastern Alps are densely populated and industrialised, anthropogenic signals are recorded alongside seismic events. We identify these anthropogenic signals based on their typical temporal signatures and confirm their origins using satellite images. Finally, we describe in detail the characteristics of the seismicity distribution.





## 2 Data

This study focuses on seismicity which was recorded by the Swath-D network (ZS) (Heit et al., 2021). The Swath-D network complemented and densified the larger-scale AlpArray backbone network (Hetényi et al., 2018) in the Eastern Alps. It formed a temporary array of broadband seismic stations and recorded during the time period from late 2017 to late 2019.

Swath-D had a total of 151 stations initially, with an aperture of around 15 km, establishing an unprecedented station coverage for the region. By the end of 2018, ten additional stations were added, extending the network to the east (Schlömer et al., 2022). Data from these stations were not used in this study to ensure a constant spatial extent for the two-year period. For completeness, all publicly available broadband stations within approximately 50 km of the Swath-D network that were active in that period were also included in this study. These comprise stations from OGS (Istituto Nazionale di Oceanografia e di Geofisica Sperimentale (OGS), 2016), LMU (Department Of Earth And Environmental Sciences, Geophysical Observatory, University Of München, 2001), SED (Swiss Seismological Service (SED) at ETH Zurich, 1983), INGV (INGV Seismological Data Centre, 2006) and ZAMG (ZAMG-Zentralanstalt Für Meterologie Und Geodynamik, 1987), as well as stations from the AlpArray backbone network (Hetényi et al., 2018). Figure 1 shows a map of the distribution of stations in the Swath-D network, as well as of all other stations used in this study.

## 3 Method

In the following, we describe our event detection and location workflow in more detail. It is sketched in Fig. 2 and consists of three main components: (1) compilation of a preliminary event list and clustering; (2) GPU-based template matching for detecting small events, and (3) cluster-based arrival time assignment & pick refinement.

### 3.1 Compilation of a preliminary event list and clustering

A basic requirement for efficient event search by template matching is a robust list of pre-selected events whose waveforms define the templates. Several agencies provide earthquake catalogues that cover at least part of the study region in the Eastern Alps (Istituto Nazionale di Oceanografia e di Geofisica Sperimentale (OGS), 2016; INGV Seismological Data Centre, 2006; ZAMG-Zentralanstalt Für Meterologie Und Geodynamik, 1987; Swiss Seismological Service (SED) at ETH Zurich, 1983). We merge these catalogues and remove multiple entries by imposing a minimum interevent time of 15 seconds or a minimum distance of 50 km. Only events with a maximum distance of 50 km to the nearest station within the Swath-D network are considered. We found 3, 455 unique events that meet these criteria.

Additionally, we apply the energy-based detection algorithm *Lassie* (Heimann et al., 2017) to the entire continuous waveform database of the Swath-D experiment in order to exploit the increased station density compared to the permanent networks in the area. This also compensates for the heterogeneous sensitivity of the available catalogues in the study region and identifies previously undetected earthquakes particularly in the central part of Swath-D. The method involves stacking the energy



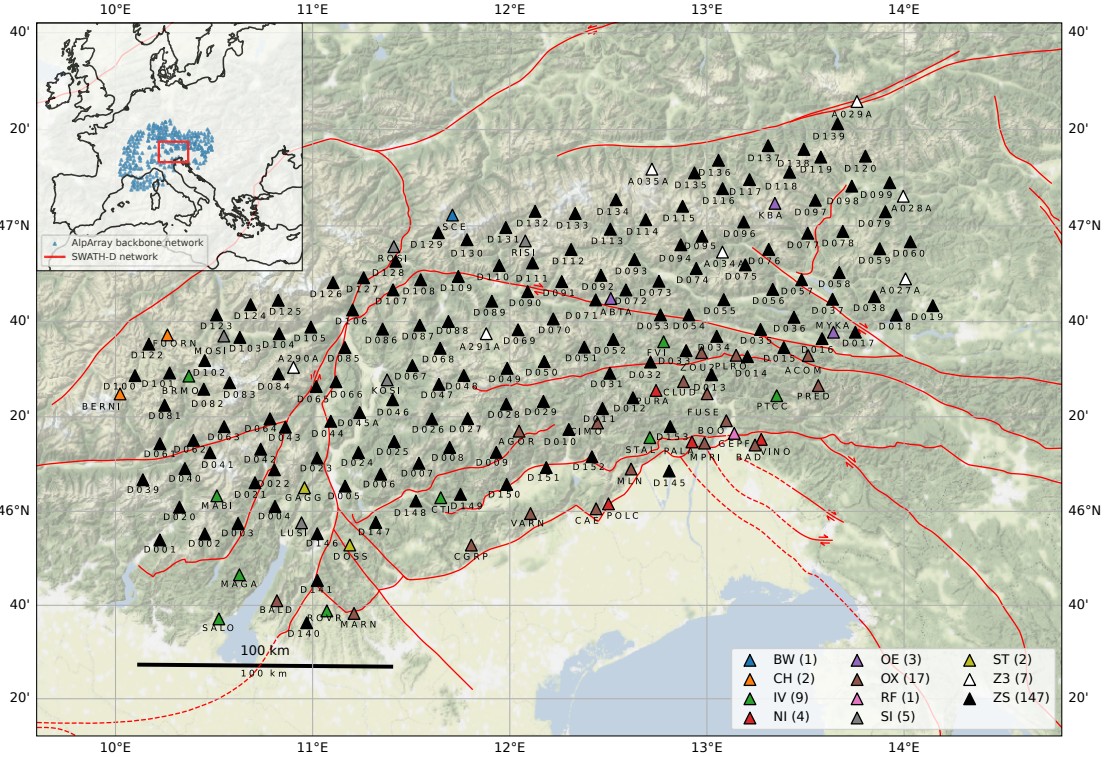

**Figure 1.** Map of the study area in the Eastern Alps showing the distribution of the stations used. The colours refer to the networks. Network BW is managed by the Department Of Earth And Environmental Sciences, Geophysical Observatory, University Of München (2001), network CH is managed by the Swiss Seismological Service (SED) at ETH Zurich (1983), networks IV, NI (in collaboration with the OGS), RF, SI, and ST are managed by the INGV Seismological Data Centre (2006), network OE is managed by the ZAMG-Zentralanstalt Für Meterologie Und Geodynamik (1987), network OX is managed by Istituto Nazionale di Oceanografia e di Geofisica Sperimentale (OGS) (2016) in collaboration with the INGV. Map background by Stamen Design under CC BY 3.0.

functions of different stations assuming a predefined grid of potential hypocentral locations and origin times. This allows to detect the source of coherent seismic signals, of which the origin time corresponds to the temporal coordinate of the maximum, and the spatial coordinates of the maximum can be used to infer its approximate location. The extent of the search region is $360\,\mathrm{km}$ by $240\,\mathrm{km}$ in the longitudinal and latitudinal directions with $5\,\mathrm{km}$ node spacing. The source depth is fixed to $15\,\mathrm{km}$. By default, data from all stations contribute to the characteristic function for each grid node. This decreases the sensitivity for small events, because their signals will be registered only by stations close to the event epicentre, and by including too many stations, these signals will be diluted. We therefore implemented a new class for the weighting of the individual stations, so that a maximum number of contributing stations around each node can be fixed. Of the $3,511$ events that are detected using Lassie, the majority is found to exist in at least one of the public catalogues. From the remaining detections, $592$ events can be located. $306$ of these remaining events are classified as anthropogenic noise (see Sect. 4 for details on the classification),



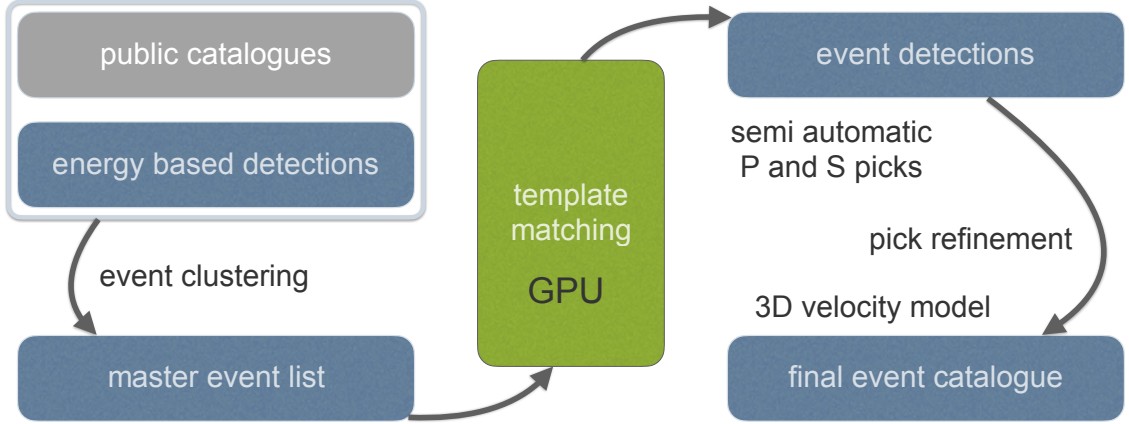

**Figure 2.** Schematic overview of the workflow and methods applied in this study.

leaving 286 potential new earthquake detections. More details, including an example of an event detection using this method can be found in Sect. A1 in the Appendix.

The catalogue earthquakes and the additionally detected events are then combined, and waveforms are cross-correlated to group events into clusters. Two events are connected if the event waveforms are similar (i.e. the cross-correlation coefficient exceeds 0.7). The similarity is measured for a 10 s time window that includes both the *P*-phase and *S*-phase arrival on the vertical channel. Within each cluster, the event that has the highest number of connections is selected as master event.

In the specific case of the Swath-D data, it turns out that the signal to noise ratio is poor for many low-magnitude events, and in some cases coherent noise leads to erroneously merging groups of events into a common cluster without being actually similar events. To solve this problem, we apply the Louvain method (Blondel et al., 2008) to find the partition of the cluster into sub-clusters (also communities) that maximises the modularity. This is a measure of the amount of connections within a sub-cluster compared to the connections to the rest of the network (Newman, 2006). If the modularity is below 0.5, we consider the cluster well connected as a whole, and leave it as it is. If the modularity is 0.5 or higher, we accept the partition, and split the cluster into sub-clusters, each with an individual master event that has the highest number of connections within the sub-cluster. This results in a list of 2,036 events to be used as master events for template matching.

### 3.2 GPU-based template matching for detecting low-magnitude events

*Preprocessing*

Template matching in seismology is usually performed by using either single phases (e.g. Ross et al., 2019), or the entire event waveforms as templates (e.g. Beaucé et al., 2019). Using the *P*- and *S*-phases separately has the advantage that the *S-P* travel-time is not constrained, and therefore events can be detected further away from the master events. On the other hand,



using shorter template waveforms massively increases the number of false triggers for prevalent weak signals as is the case for
the Swath-D network. We therefore choose to use the entire event waveform.

For each of the $2,036$ master events, we extract template waveforms of $10\,\text{s}$ length to assure that the *P*- and *S*-phase are
both included for local events. The template window starts approximately two seconds before the *P*-wave onset and is cal-
culated by assuming a straight ray-path and constant *P*-wave velocity. All waveforms are downsampled to $50\,\text{Hz}$ to decrease
computation time. We apply a bandpass-filter between $2-8\,\text{Hz}$. This suppresses high frequency noise recorded by some of
the temporary Swath-D stations, and allows slightly more variation in the event waveforms of the detected events compared
to using a wider range of frequencies. Note that we do use a wider frequency band for picking to improve the accuracy (see
Sect. 3.3). For each master event, template waveforms are extracted on the vertical component of the 15 stations closest to
the event hypocentre. Additionally, a STA/LTA trigger is applied to the waveforms to make sure that a transient signal occurs
within the template window. The STA window ($0.1\,\text{s}$) should have at least 5 times the amount of energy as the LTA window
($1\,\text{s}$). Template waveforms that do not meet this criterion are dismissed. This results in $28,207$ template waveforms, averaging
14 template waveforms per event.

### *Template Matching*

For each template waveform we then search for similar events. This approach is computationally demanding, in particular
for distributed seismic networks, because it involves for each component cross-correlation with the corresponding continuous
waveform data. To make it feasible for the Swath-D station configuration, we decided to implement our own GPU-based tem-
plate matching code. We achieve a major performance increase by reformulating the cross-correlation as a matrix multiplication
in the frequency-domain using *CuPy*, a Python library for GPU accelerated computing (Okuta et al., 2017). Additionally, the
number of I/O operations is reduced to a minimum by loading each continuous data trace only once. Using a single *Nvidia
GV100* graphics processing unit (GPU) with $8,192$ cores, our implementation handles the continuous Swath-D data in approx-
imately 14 days.

A peak search algorithm sorts all peaks in the cross-correlation function, and dismisses any peak that is within $10\,\text{s}$ of a
larger peak. A match is triggered when the cross-correlation coefficient, *cc*, exceeds $0.5$ on three different stations within a $5\,\text{s}$
time window, using templates that were extracted for the same master event. The $5\,\text{s}$ time window corresponds to the estimated
origin time of the detections. This estimate is obtained by adding the cross-correlation shift, the start time of the data trace, and
the origin time of the master event, and subtracting the start time of the template window.

This method yields $15,155$ event detections including self-detections of the master events, a 7-fold increase with respect to
the number of master events, or a 4-fold increase with respect to the combined public catalogues and energy based detections.
At this stage, the detections may include anthropogenic signals such as quarry blasts, which were partially omitted from the
public earthquake catalogues.



**Figure 3.** Example of template matching results for a single template on one station. The upper left trace in orange is the *template* waveform. The other traces are *detections* sorted by the cross-correlation coefficients starting with the *self-detection* (of the template trace within the continuous data) in blue. The other detections are coloured red if they can be associated to an event in any of the public catalogues, or green if they can not be associated to any known event. The maximum amplitude of each signal is indicated as a factor of the maximum amplitude of the template waveform. It can be observed that all signals with equal or larger amplitude than the template signal appear in one of the public catalogues, wheres all newly detected signals have lower amplitudes than the template.

## 3.3 Cluster-based semi-automatic P & S arrival time assignment & pick refinement

Through template matching, the detected events are implicitly clustered into event families. These contain all events that were detected using templates from the same master event. The events within one family have similar location because they have both similar waveforms for at least three stations as well as similar relative travel-times for these stations. This means that we





roughly know when to expect phase arrivals for the events within one family, as long as we have picks for at least one event. In this section, we describe our procedure for picking the *P*- & *S*-arrival times of all detected events, starting with one hand-picked master event for each event family.

Firstly, we hand-pick *P*- and one *S*-phase onsets on the 10 closest stations for all master events. This gives us $23,426$ picks, an average of 11 picks per event. These picks are referred to as *master picks* in the rest of this section. For all other detected events within the cluster, we estimate the phase arrivals by adding the travel-times from the master event to the origin time estimate. We then define a window of $3\,\mathrm{s}$ around this first guess, that we refer to as the *predicted phase-window*.

Secondly, we preprocess the waveforms phase-dependently. A $1-20\,\mathrm{Hz}$ bandpass-filter is applied for *P*-phases, and a $1-25\,\mathrm{Hz}$ is used for the *S*-phases. These frequency bands are wider than the one used for template matching (see Sect. 3.2) to improve the accuracy of the picks. A short-term-average to long-term-average ratio (STA/LTA) is calculated for the predicted phase-window, with an STA length of $0.2\,\mathrm{s}$ and an LTA length of $0.6\,\mathrm{s}$ for *P*-phases and $0.8\,\mathrm{s}$ for *S*-phases, respectively. The STA window runs ahead of the current sample, whereas the LTA window is behind the current sample, as by the definition of Earle and Shearer (1994). In this way, the maximum value aligns with the phase onset. Secondly, the predicted phase-window is cross-correlated with the corresponding master pick. Time windows of $0.5\,\mathrm{s}$ and $1.5\,\mathrm{s}$ around the master pick are used for *P*- and *S*-phases, respectively, both starting $0.2\,\mathrm{s}$ before the actual pick. A new trigger function is defined as the point-wise multiplication of the normalised STA/LTA ratio and the normalised cross-correlation function. The final pick time is set to the maximum of the trigger function, with the possibility of setting individual thresholds for the cross-correlation function and the STA/LTA at this point. Approximately 65-thousand picks are made using this method. An example is shown in Fig. A2 in the appendix.

Thirdly, we use a classic STA/LTA trigger to pick phases on additional stations. There are two reasons for this. The first reason is that we considered only 10 stations for hand-picking, although for larger events many more stations show phase arrivals that could be picked. The second reason is that the station availability for the network changes over time, so station-wise cross-correlation of phase windows is not possible for all event pairs. The STA/LTA trigger is applied after a first relocation iteration, so that phase windows can be estimated more accurately, yielding an additional 30-thousand picks.

This combined arrival time picking procedure provides roughly 95-thousand new *P*- & *S*-picks for a total of about 9-thousand events, originally starting with a small subset of about two thousand master events. These numbers will be slightly reduced during the location procedure, where picks that produce anomalous residuals are dismissed and a minimum number of picks per event is required.

Since we have used a mixture of three different picking techniques, we finally adopt a method proposed by Shearer (1997) to homogenise our set of picks. The method combines, for each channel and phase, cross correlation-based differential travel times $dt_{ij}$ and absolute time picks $t_i$ within each template family, and solves for an improved set of adjusted time picks $T_i$. We apply this technique iteratively, and obtain a more consistent final set of *P*- & *S*-arrival time picks. For more details on the





method and for data examples, we refer to Appendix A3.

The determined P and S arrival times are then inverted for event locations, as described in the following section. We use the
probabilistic NonLinLoc software (Lomax et al., 2000) and a recent 3-D velocity model for both P and S phases based on local
seismic tomography by Jozi Najafabadi et al. (2022), which fully covers the study region.

## 4 Results & discussion: Seismicity of the Eastern Alps

On the basis of our preliminary event list ($2,036$ events, see Sect. 3.1 for details), we detected $15,155$ events using template
matching. Each detection consists of minimum three independent, simultaneous measurements on three different stations as
explained in Sect. 3.2. By applying our specific semi-automatic picking workflow (Sect. 3.3), we are able to successfully pick
and relocate $7,756$ events (Fig. 4).

For an event to be located, we require a minimum number of 6 picks, with at least one station that has both a *P*- and *S*-pick
in order to better constrain event depth. The maximum allowed weighted residuals RMS after inverting for the hypocentre is
$250\,\mathrm{ms}$, where the residuals are subjected to a station correction term based on the median residual within the template family.
Our final catalogue is based on a total of $93,576$ picks, averaging at 12 picks per event.

### 4.1 Event classification

To first differentiate between natural seismicity and anthropogenic events, we analyse the temporal patterns of each individual
template family. Natural seismicity is expected to be randomly distributed in time, whereas anthropogenic noise usually occurs
at daytime, either within a limited time window, or at fixed times. Also, the number of anthropogenic noise events is much
higher during weekdays compared to weekends. Histograms of these temporal features are inspected manually for all template
families containing three or more events. In case an anthropogenic signature is suspected, we consult satellite imagery to look
for quarries or construction sites. This leads to the identification of $1,659$ events related to two dozens of anthropogenic noise
sources. In Fig. 4, their distribution is indicated by orange symbols. The distinct temporal patterns of both classes are shown in
Fig. 5, also revealing a slightly increased sensitivity to natural seismic events during nighttime and on weekend days.

### 4.2 Magnitude distribution

After removing the anthropogenic signals from our catalogue, $6,053$ events remain which we classify as tectonic earthquakes
(blue circles in Fig. 4). Their magnitude and temporal distributions are shown in Fig. 6 and 7. As expected, template match-
ing particularly increases the number of low magnitude events. The smallest located earthquakes have a magnitude of about
$M_l = -0.5$, the magnitude of completeness is $M_c \approx 1.0$. The estimated $b-value$ (Gutenberg and Richter, 1944) is $0.93$ and



**Figure 4.** Map of the distribution of events found in this study, classified as earthquakes (blue dots) and anthropogenic events (orange dots). Major faults from Schmid et al. (2004) are indicated with red lines. The abbreviations are SEMP: Salzach-Enns-Mariazell-Puchberg Fault, PGF: Pustertal-Gailtal Fault, BF: Brenner Fault, GF: Giudicarie Fault, VF: Valsugana Fault, KF: Katschberg Fault, MF: Mölltal Fault, FSF: Fella-Sava Fault, MO-FR: Montello-Friuli thrust belt. Tectonic units are named as in Reiter et al. (2018), TW: Tauern Window, DI: Dolomite Indenter, FR: Friuli, GL: Giudicarie-Lessini region, EA: Engadine Alps, TM: Texel Group and Meran-Passeier area, SA: Stubai Alps. Symbols $A - F$ mark the locations of earthquake sequences indicated in Fig. 7. Map background by Stamen Design under CC BY 3.0.

slightly higher than the value obtained for the limited group of master events ($b = 0.84$; see Appendix A4 for the magnitude calculation method).



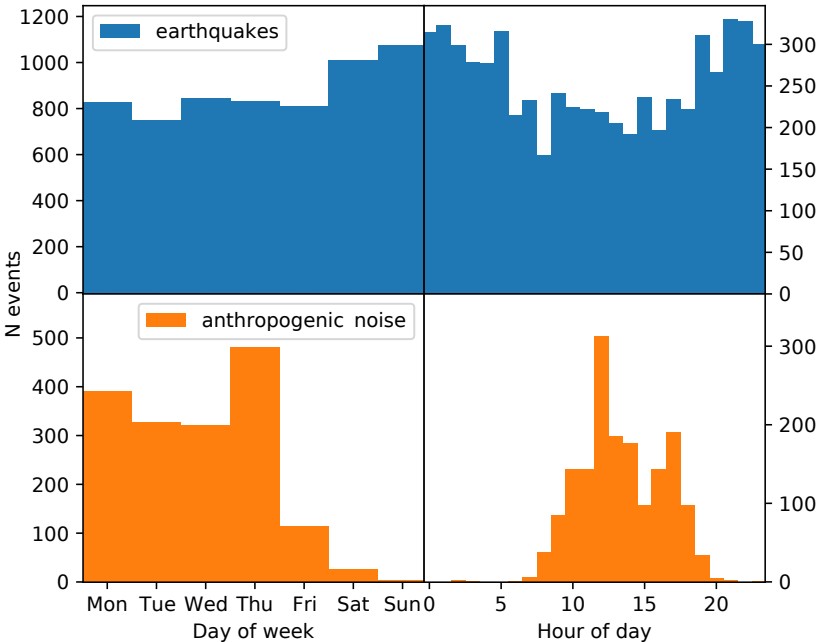

**Figure 5.** Histograms of the weekday (left panels) and hour of day (right panels) for the events classified as earthquakes (upper panels) and anthropogenic noise (lower panels).

## 4.3 Temporal patterns in seismicity

Throughout the recording period, we observe a fairly constant background seismicity rate of about twenty events per day (Fig. 7). This mode is alternated with short episodes of intense seismic activity, which become particularly evident by the template matching. There are six occurrences where the daily detection rate exceeds one-hundred events per day. These sequences and

their locations are marked by symbols $A - F$ in Fig. 4 and 7. The first is on November 3rd, 2017 ($A$). This corresponds to a seismicity cluster in the Stubai Alps, south of the city of Innsbruck. The largest earthquake in this sequence has a magnitude of $M_l = 3.6$. The second and largest peak is on February 25th, 2018 ($B$). This cluster is located north of the village of Cimolais, on the Friuli-Veneto border. It contains a magnitude $M_l = 3.6$ event and a magnitude $M_l = 3.4$ event. The third is on April 25th, 2018 ($C$), with a maximum magnitude of $M_l = 2.6$ in the Münstertal valley on the Swiss-Italian border. The fourth is on

August 11th, 2018 ($D$). This sequence is located south of the town of Tolmezzo, and is contains one event with a magnitude of $M_l = 3.4$. The fifth is on February 1st, 2019 ($E$). This sequence in the Friuli region west of the village of Chievolis has a maximum magnitude of $M_l = 1.2$. The sixth and last is on February 9th, 2019 ($F$) in the Suldental valley on the Swiss-Italian border. This sequence has a maximum magnitude of $M_l = 1.9$. As an example, the relocated events of this cluster are shown in Fig. A9.



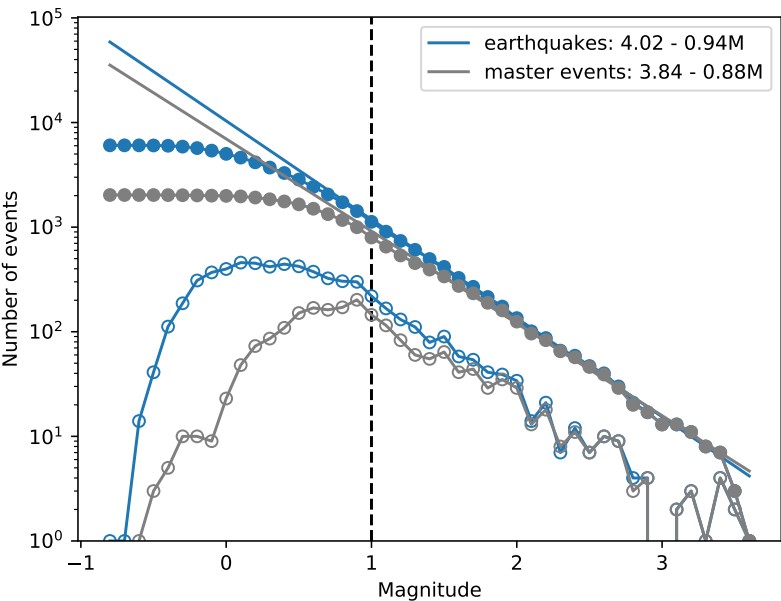

**Figure 6.** Gutenberg-Richter plot showing the distribution of (relative) magnitudes in the input catalogue (grey) and the final earthquake catalogue (blue) (Gutenberg and Richter, 1944)

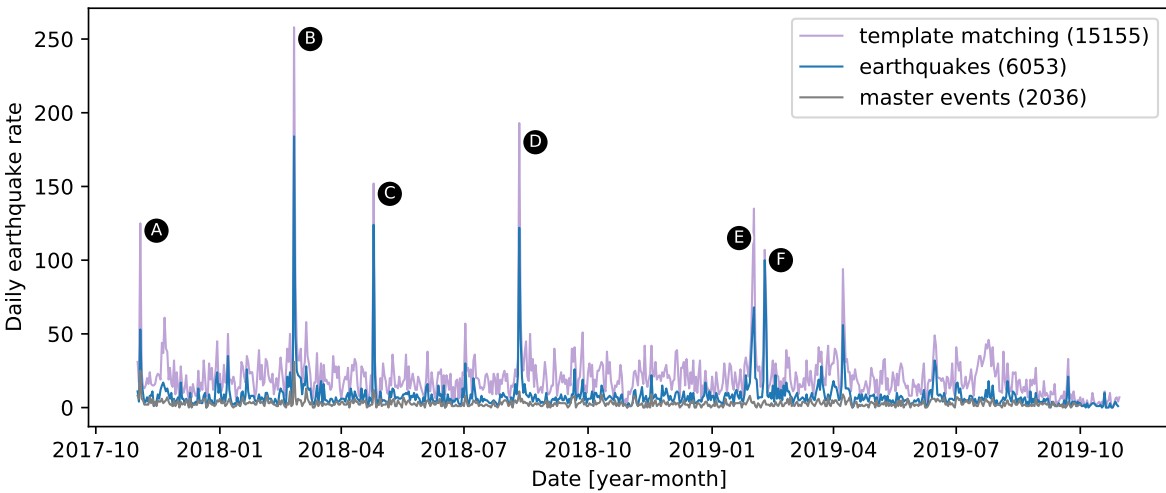

**Figure 7.** Daily earthquake rate in the initial input catalogue (grey), the template matching detections (purple), and the final relocated catalogue (blue). Sequences that exceed a daily rate of 100 events are indicated with symbols $A - F$ in Fig. 4.



## 4.4 Spatial distribution of seismicity

The epicentral seismicity distribution from this study, which is restricted to the recording period of the Swath-D network in the years 2017-2019 (Fig. 4 and 8), largely corresponds to the long-term behaviour of $M_l > 2.0$ earthquakes obtained in earlier studies (Slejko et al., 1998; Reiter et al., 2018). Seismicity is mostly shallow, with 98% of the seismic events occurring within the upper $20\,\mathrm{km}$ of the crust (see the depth histogram enclosed in Fig. 4). This is consistent with previous studies in the Eastern Alps (Reinecker and Lenhardt, 1999; Viganò et al., 2015; Reiter et al., 2018; Jozi Najafabadi et al., 2021). The major part of the event hypocentres (75%) are in the band of $5\,\mathrm{km}$ to $15\,\mathrm{km}$ depth, with pronounced concentrations of earthquakes between $8\,\mathrm{km}$ and $12\,\mathrm{km}$ depth.

The spatial patterns reflect the NNW convergence of the Adriatic microplate and the European plate, with active seismic deformation occurring mostly in the thrust- and- fold belt in the South-Eastern Alps, which separates the indenter from the undeformed part of Adria (Castellarin et al., 2006; Cheloni et al., 2014; Serpelloni et al., 2016; Reiter et al., 2018; Petersen et al., 2021). Here, seismicity is most prevalent in the Montello-Friuli thrust belt in the southeastern corner of the study area (Romano et al., 2019; Bragato et al., 2021). This is by far the most seismically active area during the recording time of the Swath-D network. This area was witness to the largest instrumental earthquake in northern Italy, the devastating $M_l = 6.4$ Friuli earthquake (e.g. Slejko, 2018; Aoudia et al., 2000). The density of magnitude $M_l = 3$ events in the area is much higher here compared to the rest of the study area. To the south, the seismic activity spills over into the Po basin, following the pattern of known historical seismicity, whereas to the north, most of the seismic activity is cut off sharply by the Fella-Sava and Valsugana Faults. Seismicity to the north of these faults, within the north western part of the Dolomite indenter, is much sparser. Recent crustal tomography studies show this area is characterised by a positive anomaly of Rayleigh wave phase velocities at seismogenic depths, indicating a denser and possibly more rigid upper crustal block (Sadeghi-Bagherabadi et al., 2021; Kästle et al., 2021).

Nonetheless we are able to relocate roughly one hundred events, benefiting here particularly from the improved resolution in the central part of Swath-D network compared to the permanent station configuration. These events are generally below magnitude $M_l = 2$, except for two events close to the Fella-Sava fault. The seismicity rate increases slightly within the Tauern window further northwards, starting on the Pustertal-Gailtal fault, a segment of the dextral Periadriatic fault. Our catalogue contains roughly two hundred events in this area. From modelling GPS velocities, the Pustertal–Gailtal fault is assumed to accommodate part of the NW indention (Caporali et al., 2013). We find earthquakes that align along an almost $100\,\mathrm{km}$ long section of the Pustertal-Gailtal fault between about $12.0°\,\mathrm{E}$ and $13.0°\,\mathrm{E}$. The strike-parallel depth section in Fig. 8 (profile $EE'$) shows the seismicity here at a fairly constant depth of $5-10\,\mathrm{km}$.

Further to the east, a few events are also observed in a few clusters at variable depths between about $5-12\,\mathrm{km}$ at the Katschberg Fault and the Mölltal Fault, extending to its northwestern end at about $(13.15°\,\mathrm{E}\,|\,47.0°\,\mathrm{N})$.



Seismicity within the Tauern window is equally low in magnitude, with only a few $M_l = 2$ events. North of the Salzach-Enns-Mariazell-Puchberg Fault, the seismicity rates as well as the magnitudes increase again, but this area lies outside of the Swath-D area and is therefore not within the scope of this paper.

Elevated levels of seismicity are also observed to the west of the indenter boundary, primarily in the Giudicarie-Lessini region and the Engadine Alps, where the seismicity seems to be homogeneously spread without following well defined patterns. These areas show frequent $M_l = 2$ events. Further north in the Texel group and Meran-Passeier area, where the $M_l = 5.3$ in 2001 occurred (Viganò et al., 2015; Reiter et al., 2018), and the Stubai Alps, seismicity is more focused in a few active clusters. Seismicity rates remain high and magnitudes elevated in the Inntal region on the northernmost detection limit.

We provide two animations in the online supplement that illustrate the 3-D distribution of the here produced Swath-D seismicity catalogue completely. They allow to visualise the seismicity in continuous north-south and east-west depth slices. Fig. 8 shows depth sections along four profiles in different parts of the study region, which provide a more detailed view of the fault activity at depth. They highlight the significant variations of seismic activity from west to east and illuminate the
varying complexity of the seismically active structures. In particular, the easternmost profile D in the thrust- and- fold belt in the South-Eastern Alps at $\sim 13°$ E resolves a divergence of dip angles and directions which reflects the intricate interaction of Alpine and Dinaric tectonics (e.g. Bressan et al., 2021).

### 4.5 Location uncertainty

Location uncertainty can be estimated from the 68% confidence ellipsoids derived from the probability density functions
(PDFs) in NonLinLoc. We observe that the semi-major axes of the ellipsoids are predominantly vertical, meaning that the horizontal coordinates of the seismicity are better resolved than its depth. Mean location errors are in the order of $300\,\text{m}$. Standard deviations of the normally distributed P and S arrival time residuals for all located events are about $0.07\,\text{s}$ for P picks and $0.10\,\text{s}$ for S picks, respectively, and are shown in Fig. A6 in the appendix. As an additional appraisal of the location uncertainty, we studied the spatial distribution within event families by calculating the distance of the events relative to the
master events. Dependent on the threshold of waveform similarity, the number of events rapidly decreases with interevent distance (Fig. A7). For highly similar event pairs with waveform similarity $cc > 0.9$, the horizontal event distances are mostly less than $1\,\text{km}$, and slightly larger for the vertical distance ($1.5\,\text{km}$). These distances represent an upper bound on the location uncertainties. In Fig. A8 we analyse the direction of the relative locations with respect to the master events, and show that no azimuthal bias can be observed. Figure A9 shows an example of an event cluster with 239 events, detected and picked based
on 7 master events only. This demonstrates the potential of resolving individual structures using our workflow.

## 5 Conclusions

We have presented an efficient workflow particularly designed for detection, phase picking, arrival time pick refinement and location of small earthquakes and apply it to the dense Swath-D seismic monitoring network in the Eastern Alps. Although the





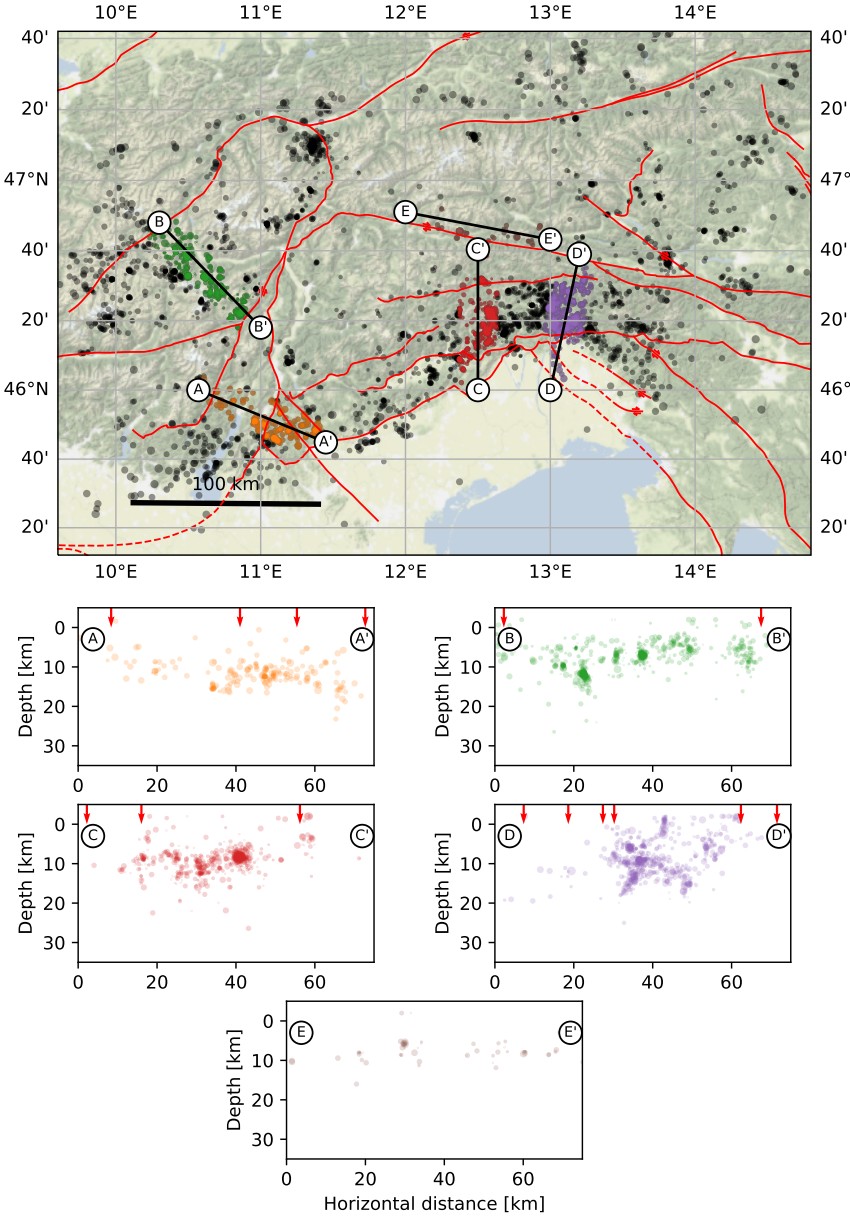

**Figure 8.** Four exemplary cross-sections of the seismicity in the area. The horizontal axes of the sections are indicated on the map in the upper panel. The colours are solely to indicate which events are used in the sections. Vertical red arrows mark the intersections of the cross-section with fault traces shown on the map. Map background by Stamen Design under CC BY 3.0.

region is monitored routinely by several national agencies that provide high-quality seismic event catalogues, our workflow, in combination with the integration of the new Swath-D seismological data, increases the number of detected events. The main components of the workflow are a systematic GPU-based template search to detect low-magnitude events and a semi-automatic





picking routine to enhance the quality of P & S arrival times and their consistency. This facilitates for the first time a coherent and detailed location of the seismicity in the study region, based on the recent 3-D velocity model by Jozi Najafabadi et al. (2022).

Our catalogue includes $6,053$ events for a time period of 24 months in 2017 to 2019, with magnitudes in the range $-0.5 \leq M_l \leq 3.7$ and a magnitude of completeness of $M_c \approx 1.0$. The earthquake distribution retrieves the main patterns of seismicity previously found by long-term, but coarser seismic monitoring of the region. Earthquake depths are strongly focused in the upper crust within a relative narrow band of $7 - 15$ km depth. The improved resolution of this study allows to detect and locate additional, mostly weak events, a part of them pointing to small, but active deformation at upper crust level in the Dolomite indenter, along the Pustertal-Gailtal Fault and in the Tauern window, and the newly revealed clusters overall better illuminate the fault structures at depth.

The here presented image of micro seismicity provides an important new and extended basis for future detailed studies of seismicity and tectonic structures in the Eastern Alps.

*Data availability.* We plan to publish the earthquake catalogue on the project website with a DOI (http://dataservices.gfz-potsdam.de/4dmb/) after the manuscript has been accepted. On request, it will of course be available to the reviewers.

*Video supplement.* We plan to publish two videos with moving North-South and East-West cross-sections along with our catalogue on the project website with a DOI (http://dataservices.gfz-potsdam.de/4dmb/) after the manuscript has been accepted. If necessary, they will of course be available to the reviewers.

*Team list.* For the complete member list of the AlpArray-Swath-D working group, please visit https://doi.org/10.14470/MF7562601148.





# Appendix A

## A1  Lassie waveform-based event detection

Figure A1 shows a seismic event detection using Pyrocko's Lassie detector (Heimann et al., 2017). This event was detected within the Dolomite indenter (at $11.57° \text{E} \mid 46.76° \text{N}$) and could not be linked to any event from any of the public catalogues. Because the Dolomite indenter and the Tauern window have relatively low seismicity rates, the public networks have fewer permanent recording stations in these areas. Therefore, these areas benefit most from the deployment of the Swath-D network. This example shows that it is possible to detect new events using fully automated methods based on the Swath-D data, demonstrating an increased resolution in this part of the area.

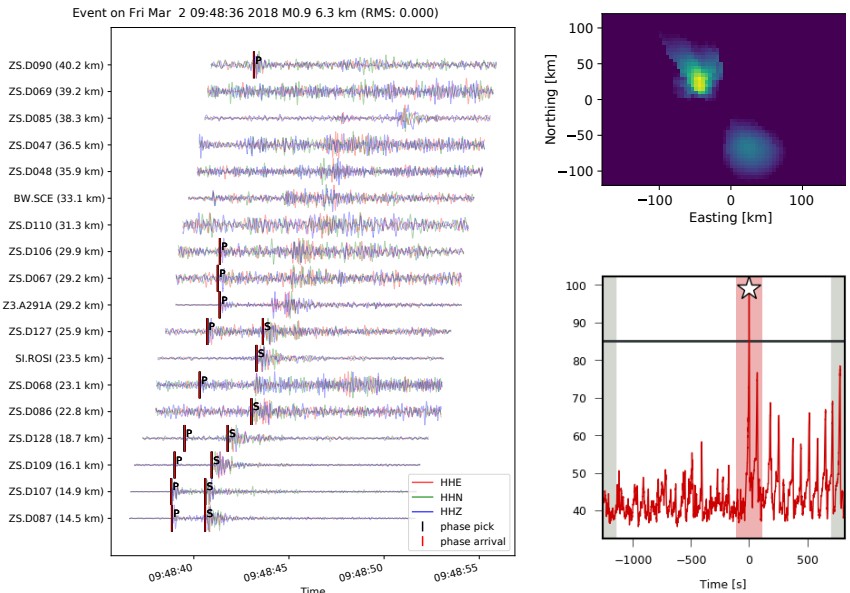

**Figure A1.** Example of a seismic event detected using Lassie (Heimann et al., 2017). The event is located within the Dolomite indenter at $11.57° \text{E} \mid 46.76° \text{N}$ at a depth of $6.3 \, km$ and occurred on march second, 2018 at $9 : 48 : 36 \, \text{AM UTC}$. The left panel shows the event waveforms, sorted by hypocentral distance. The upper right panel shows the image function in space, the lower right panel shows the characteristic function in time.





## A2 Illustration & examples of cluster-based P & S arrival time picking

The methods used for phase-picking of the detected seismicity are explained in Sect. 3.3, but a few additional, illustrative

figures are shown in this section. Figure A2 shows an example of our picking method, where a hand-picked phase from a master event is transferred to a detected event using the cross-correlation function, and an STA/LTA filter. In this example, using only the cross-correlation function (to correlate the hand-picked phase with the expected phase-window of the detected event), would lead to a false pick. The normalisation of the cross-correlation function causes the amplification of noise in the time window before the actual phase arrival. By multiplying the cross-correlation function with the normalised STA/LTA filter,

we suppress the pre-arrival noise and ensure that a transient signal is present where we finally set our pick.

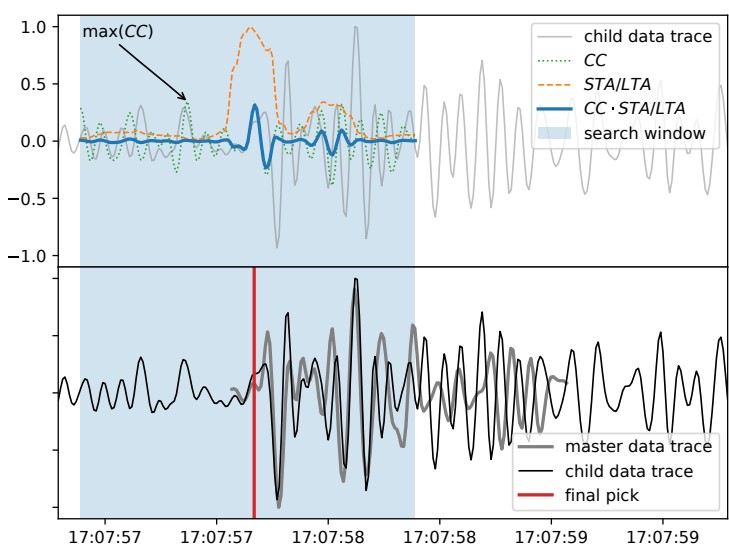

**Figure A2.** Example of a manual pick transferred to a detected event using the product of the cross-correlation function, and an STA/LTA filter. In this case, using only the cross-correlation would result in a wrong pick. By multiplying both functions, we get a more accurate result. The upper panel shows the cross-correlation function, the STA/LTA filter, and the product of both functions. The lower panel shows the resulting pick and the data trace for the detected event, as well as the master phase used for the cross-correlation.

Another example of our picking method is presented in Fig. A3. This figure shows an example of a master event that is (mainly) hand-picked, and an event detected using that master event that is picked automatically using the methods described in Sect. 3.3.



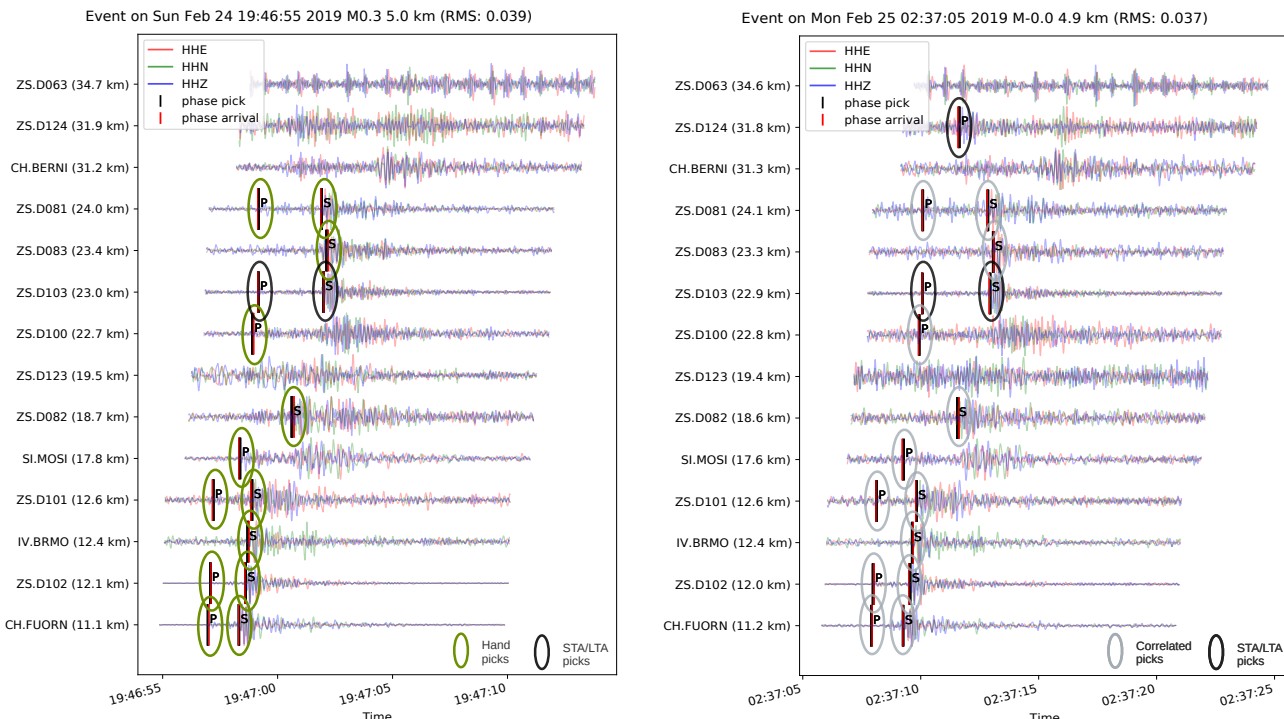

**Figure A3.** Example of event waveforms for an event used as a template (left panel), and an event that was detected using this template event (right panel). The master event is hand-picked, whereas the picks for the detected event are correlation-based (see Sect. 3.3 for details). Both events contain a few additional STA/LTA picks. The station waveforms are sorted by the hypocentral distance.



**A3   Arrival time pick optimisation**

*P-* & *S*-arrival time picks for events within each cluster are refined adopting and slightly modifying a method by Shearer (1997).
For the N events within a family ($N > 2$), we solve a set of linear equations (equation A1) for the $N$ adjusted arrival times $T_i$,
separately for each phase, either *P-* or *S*-phase, and each station. To avoid numerical problems during the inversion, all time
values are taken relative to the event origins.

$$
\begin{pmatrix}
t_1 \\
t_2 \\
t_3 \\
. \\
t_N \\
wcc_{12}\,dt_{12} \\
wcc_{13}\,dt_{13} \\
. \\
wcc_{(N-1)N}\,dt_{(N-1)N}
\end{pmatrix}
=
\begin{pmatrix}
1 & 0 & 0 & . & 0 \\
0 & 1 & 0 & . & 0 \\
0 & 0 & 1 & . & 0 \\
. & . & . & . & . \\
0 & 0 & 0 & . & 1 \\
wcc_{12} & -wcc_{12} & 0 & . & 0 \\
wcc_{13} & 0 & -wcc_{13} & . & 0 \\
. & . & . & . & . \\
0 & 0 & 0 & wcc_{(N-1)N} & -wcc_{(N-1)N}
\end{pmatrix}
\begin{pmatrix}
T_1 \\
T_2 \\
T_3 \\
. \\
T_{N-1} \\
T_N
\end{pmatrix}
\tag{A1}
$$

where $t_i$ are the preliminary time picks relative to the event origin (i.e. travel-time), $dt_{ij}$ are the correlation-based differential
travel times (cross-correlation lag time minus the difference in origin times $i$ and $j$), and $wcc_{ij}$ are weighting factors based on
the values of the normalised cross correlation coefficient $cc_{ij}$ for the event pairs $(i, j)$. Here, we use $wcc_{ij} = 1$ if $cc_{ij} > 0.7$
and 0 otherwise. The vector $T_i$ will then contain the updated time picks.

For each event family, station and seismic phase (both *P-* and *S*), we repeat the inversion for a maximum of 15 times, or until
the solution converges, and obtain collectively a more consistent set of *P-* & *S*-arrival time picks. The effect of the method is
illustrated exemplarily for one event family and one station in Fig. A4.


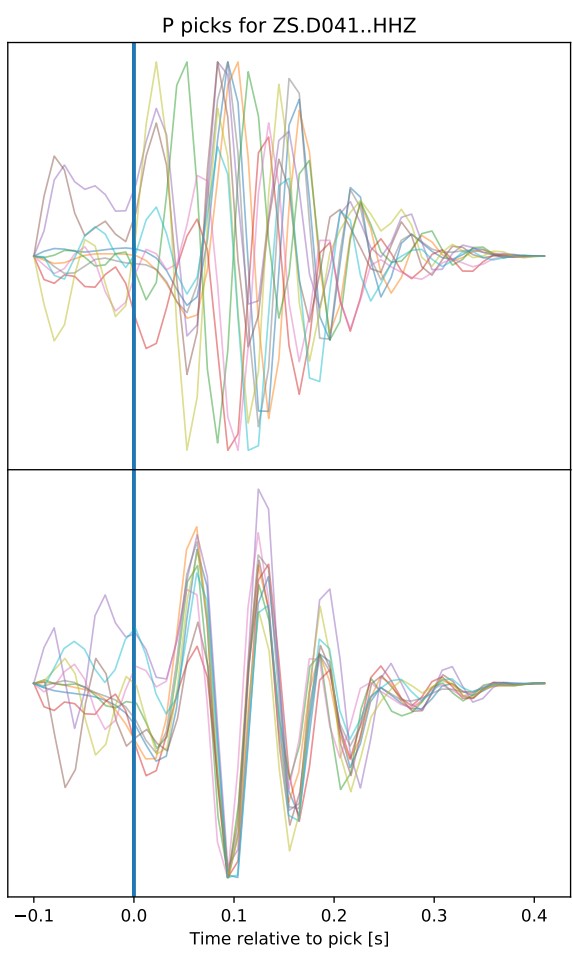

**Figure A4.** Vertical seismograms for detected events in a template family for one station, aligned on the *P*-pick time before (upper panel) and after (lower panel) applying the method described in Shearer (1997). Note the greatly improved consistency of the phase onsets in the lower panel, equivalent to an improved accuracy of the P- arrival time picks.





## A4  Calculation of event magnitudes

Event magnitudes for detected events are computed relative to the magnitudes in the publicly available event catalogues, based on peak *S*-wave amplitude ratios. For each station, peak *S*-wave amplitudes are determined for all events with known magnitudes recorded by the station. A function (equation A2) is fitted to these amplitudes that can later be used to calculate
local magnitudes for other events, as well as calibrate the magnitudes of the master events using the Swath-D data.

$$M_L = x_0 * (log_{10}(A) + x_3 * D^{x_2}) + x_1,$$   (A2)

where $A$ is the peak *S*-wave amplitude and $D$ is the horizontal distance from source to receiver, and $x_0$ to $x_3$ are calculated using a least-squares fit. Magnitudes in the final result are the average of local magnitudes for all of the *S*-picks available for the given event.
Figure 6 in the main paper shows the magnitude frequency distribution of the catalogue using the Gutenberg-Richter law (Gutenberg and Richter, 1944). The final catalogue has a significantly higher $b$-value compared to the subset of master events used as templates (see Sect. 3.1).



Figure A5 shows the relative magnitudes of the detected events in our final catalogue relative to the magnitude of the master events used as templates. It can be clearly observed that the cross-correlation threshold influences the maximum difference

in magnitude that can be detected. If a threshold of $0.9$ were used, the maximum difference would be restricted to one unit of magnitude, whereas a threshold of $0.5$ allows a difference of three units of magnitude. Note that the histogram in Fig. A5 should theoretically be symmetrical. The asymmetry is caused by the skewness of the magnitudes used as master events: the lower magnitude events are underrepresented. This figure therefore shows us that the majority of the detected seismicity is within the range of zero to two units of magnitude lower than the known seismicity in the region.

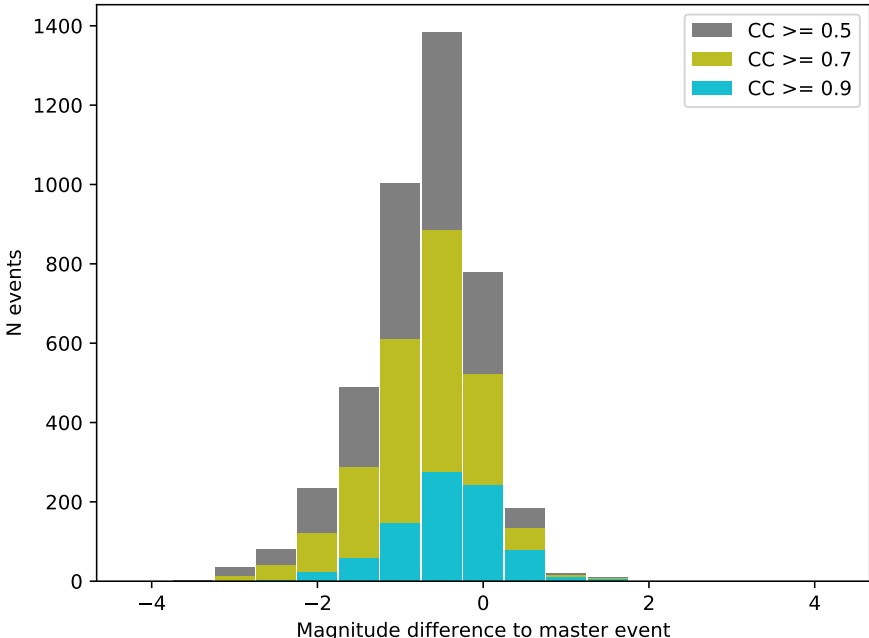

**Figure A5.** Difference in magnitude between each event and its master event.





**A5    Location uncertainty**

This section contains a few additional figures addressing the location uncertainty in our catalogue.

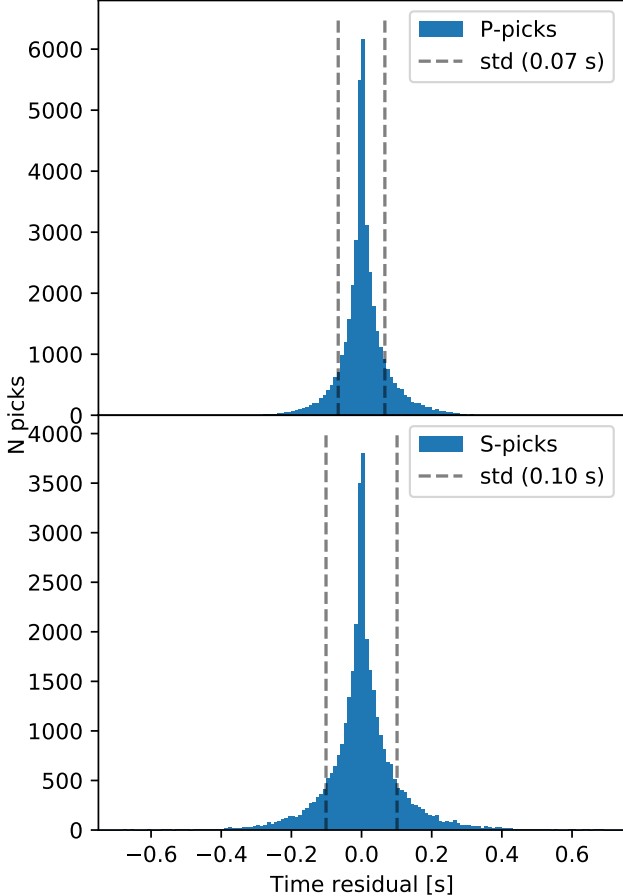

**Figure A6.** Residuals corresponding to the locations in the final earthquake catalogue for all *P*-picks (upper panel) and *S*-picks (lower panel).





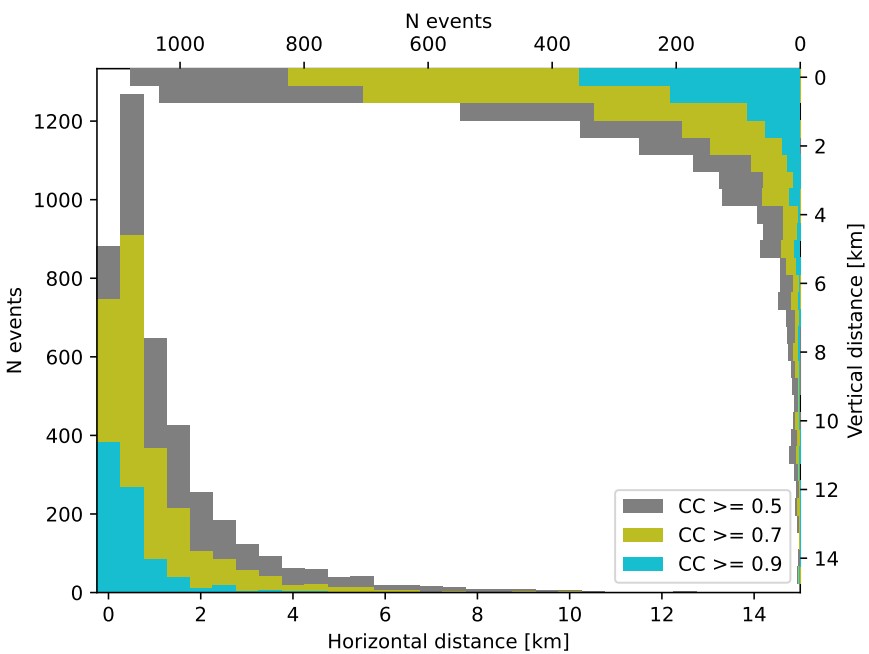

**Figure A7.** Histograms of the horizontal and vertical distance of the events in the final catalogue with respect to the master events.

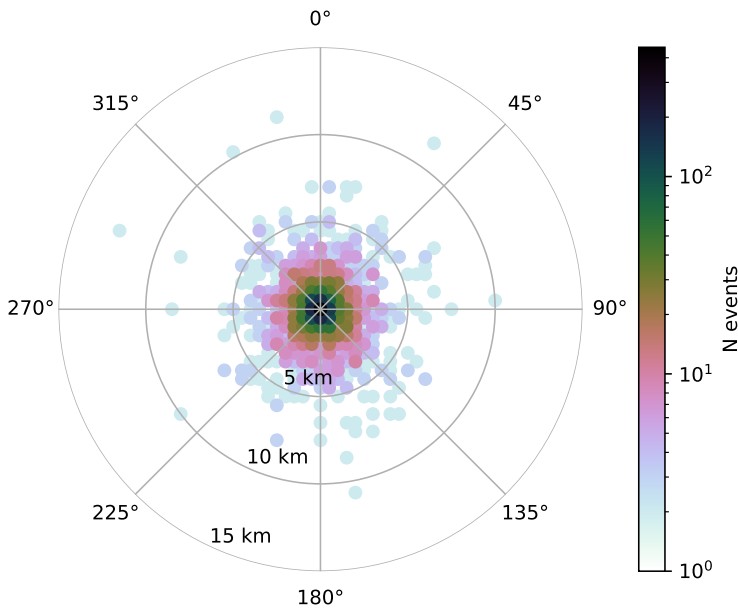

**Figure A8.** Azimuthal distribution of the events in the final catalogue with respect to the template events. The distances are three-dimensional.



**Figure A9.** Example of an event cluster consisting of 239 automatically picked and relocated earthquakes (blue circles) based on 7 master events (pink circles). This cluster (marked with the symbol $F$ in Fig. 4 and 7) is located in the Suldental valley on the Swiss-Italian border, and its main sequence took place on February 9th, 2019. Map background by Stamen Design under CC BY 3.0.



*Author contributions.* Conceptualisation and funding acquisition by JK and SC. Development of Methodology by LJH and JK. Formal analysis, investigation and software development and visualisation by LJH under supervision of JK and SC. Writing by LJH with contributions from JK and edits by CS. Project administration and fieldwork was done by the AlpArray-Swath-D working group.

*Competing interests.* The authors declare that they have no conflict of interest.

*Acknowledgements.* We thank Chistian Haberland for providing us the digital $V_P$ and $V_S$ velocity model of Jozi Najafabadi et al. (2022), and Gesa Petersen for many helpful discussions. We are also grateful to Giugliana Rossi and Alessandro Vuan for their very helpful and constructive comments on the manuscript. We thank the AlpArray Working Group for providing access to the data. A list of team members can be found at http://www.alparray.ethz.ch.

*Financial support.* This research has been supported by the German Science Foundation DFG (grant nos. KU 2484/5-2, CE 223/6-2) through the special priority programme (SPP) "Mountain Building Processes in Four Dimensions (4D-MB)".





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
