# Peer review of "A New Seismicity Catalogue of the Eastern Alps Using the Temporary Swath-D Network"

_EGUsphere, 2023_

## Referee Comment (RC1)

Review of
**A New Seismicity Catalogue of the Eastern Alps**
**Using the Temporary Swath-D Network**
by *Laurens Jan Hofman, Jörn Kummerow, Simone Cesca, and the
AlpArray-Swath-D Working Group*

In this study, the authors take advantage of a temporary, dense and relatively large seismic array (Swath-D Network) to build a new earthquake catalog and improve the current knowledge of the seismicity in the eastern and southern Alps. The authors present the results of a thorough analysis mixing automated and manual methods with which they detected and located 4 times more events than existing catalogs. The manuscript is well organized and written. I enjoyed reading about sophisticated methodology that doesn't involve machine learning. I only have minor comments about the methodology and some surprising results.

**Comments**

- Did the INGV and SED agencies use the same data set when building their catalogs? I find quite surprising that after the energy-based detection there are only 286 new events.

- Lines 104 and 296, you mention the correlation coefficient but don't give details about how many stations are involved in its computation. I think this information is important to understand what the value means.

- Line 140 "Additionally, the number of I/O operations is reduced to a minimum by loading each continuous data trace only once." I don't really understand the meaning of this statement because I don't see why one would load the data several times?

- The detection threshold is set in a very arbitrary way: How much is 0.5 in terms of standard deviations? CC detection thresholds are usually defined upon the root mean square or median absolute deviation of the CCs so that they correspond to a given p-value of false detections assuming a gaussian distribution. Based on my personal experience, 0.5 can also be very conservative. What is the motivation for not summing the CCs and applying the threshold on the summed CCs? Array techniques are all based on the idea that signal-to-noise ratio of a sum increases as $\sqrt{N}$ where $N$ is the number of traces. But by applying the detection threshold on single stations, one loses the full benefits of network information.

- Lines 166-167: For phase picking, the authors use higher frequencies for S waves than for P waves. What motivated this decision? It is unexpected since P waves carry higher frequency energy (the P-wave corner frequency is about 50% higher than that of the S wave).

- Lines 178: I got a bit lost in these explanations. We start reading about picking on the template events, then on the newly detected events, and in this paragraph the authors go back to hand picks, which I thought were only for template events? I found this paragraph hard to understand so perhaps it is worth editing it.

- Magnitude estimation. Figure A5 shows some surprising observations. The lack of knowledge of what the CC means exactly (see comment above) might partly feed this comment. For event pairs with CC > 0.9, even within the reduced frequency band of 2-8 Hz, I don't see how the magnitude difference

can be up to 2 units. Could it be caused by errors in the magnitude computation?

- Line 291: A mean location uncertainty of only 300m? Most earthquakes are located outside the network (Figure 4) so I would expect a much higher number.

- Line 297: "upper bound" on location uncertainty. I would avoid making a quantitative statement from a hand wavy argument. Is the idea that events with CC>0.9 are more-or-less exactly co-located? In this case, shouldn't it be a lower bound (true uncertainty = within-family dispersion + NLLoc uncertainties)?

- It would have been nice to have the catalog as a supplementary file. The authors do say that they are willing to share the file upon request, but this seems incompatible with anonymity.

---

## Referee Comment (RC2)

*Review Solid Earth EGUSPHERE (https://doi.org/10.5194/egusphere-2023-806)*

**A New Seismicity Catalogue of the Eastern Alps Using the Temporary Swath-D Network**

Laurens Jan Hofman, Jörn Kummerow, Simone Cesca, and the AlpArray-Swath-D Working Group

The paper from Jan Hofman et al. focuses on the development of a new semi-automatic methodology for the creation of a consistent seismic catalog in the Eastern Alps, as part of the Swath-D project. The method involves both manual and automatic procedures, as well as the GPU usage to fasten up the process. This application aimed at precise arrival pick detections for P and S phases, thus producing improved absolute seismic locations. I found the paper well written in English, with room for improvements mainly for the methodological sections. I also appreciate the authors shifting most of the methods in the Appendices for a clearer view of the main goals. Eventually, I recommend the publication of this work after revisions.

**Main General Comments:**

- To meet the FAIR publications criteria, the authors should provide the final catalog with at all the needed metadata to replicate their results. The codes used in the study should be also distributed in one or more open access repository (i.e. GitHub, GitLab, Zenodo) at authors discretion. I think that especially for methodological paper, this latter point is a must.

- In my opinion, the magnitude calculation and catalog completeness definition are a bit incomplete. First, the author should provide a clear value for the completeness consistently throughout the manuscript: the symbol ~ should not be used (i.e. P1-L3 / P9-L222/ P16-L311). Second, how is the completeness calculated? With which method? Is one or more statistical approaches involved? Finally -and most important- why not pursue a much more consistent strategy of reassessing the event local magnitude based on S-wave amplitude? Having already extracted the amplitude windows during the cross-correlation and repicking stages, it should be straightforward to apply a standard ML attenuation function and the catalog would highly benefit from it in terms of consistency and robustness, not to mention for the discussions section. One could then plot the master events initial magnitude against the recalculated one to validate a linear fit trend. I would recommend these calculations and to better validate the catalog completeness, although I leave the decision to the authors.

- Still talking about magnitudes…citing appendix A4: "*Event magnitudes for detected events are computed relative to the magnitudes in the publicly available event catalogues, based on peak S-wave amplitude ratios. For each station, peak S-wave amplitudes are determined for all events with known magnitudes recorded by the station*". In the follow up equation, though, I see the least-square fit based on master

event station magnitudes. I think this assumption could lead to biases. Indeed, the master events extracted from national bulletins are obtained with different local magnitude scales parametrization. Please provide some reasoning for your decision and try to expand the discussion in the Appendix A4.

- I would try stress a bit more the discussion over the improvement with the authors GPU support solution. Did the authors tried to benchmark their approach with other methods? Or even compare it with the serialization or standard parallelization approach? Some absolute and quantitative number would be nice to have for the reader.

- Another important yet underexplained part of the methodology is the anthropogenic / natural seismic event definition (and therefore filtering). P2-L56 *"We identify these anthropogenic signals based on their typical temporal signatures and confirm their origins using satellite images."*. Is it really based only on temporal signatures? How where the satellite images compared? The authors should provide an example of anthropogenic event detection and filtering to complement Figure 5. This stage is as important as a good picking refinement, and I think it should deserve more space in the manuscript. Did the authors adopted any criteria to stay on a more conservative scenario (i.e., losing more seismic events compared to risking having anthropogenic noise labeled as seismic events)?

- I think the author could provide an additional figure (in the supplementary) showing the x-y-z-ot uncertainty distributions of the final catalog events, to validate the average errors declaration.

**Additional comments:**

*Fig.1:*
- A differentiation with marker type (i.e., square or circles) is helpful to discern between the permanent and temporary stations. Please group them accordingly.
- Please provide a useful web page or reference for "Stamen Design" (package/tool) in the legend and therefore in the reference-list.
- The number 198 (P2-L48) correctly sum up the stations number listed in the legend. The authors, though, also mention that the SWATH-D is composed by 151 stations (P2-L32). In the figure's legend is written 147. Please double check the numbers and correct accordingly both in captions and in text.
- The Z3 and ZS temporary network listing and their respective citations are missing in the figure's caption. Please add them.
- In my opinion, displaying the fault systems is a bit off the figure's scope, as the main aim is to describe the station network. The authors correctly display and describe the fault system in Fig.4 and Fig.8, and therefore I would suggest the authors to remove them from this figure.
- I think it would be nice for the reader and for the completeness of the manuscript to provide a figure in the supplementary material with the initial

seismicity geometry (the one from the agencies bulletin) in different colors. Possibly, sided with the merged seismicity. I strongly recommend such a figure.

*Fig.2:* Is there a reasoning why the authors use different colors (blue / gray) and occasionally grouping in gray box? In the manuscript, the method part is divided in 3 main blocks, I would recommend the authors to follow that flow consistently in the figure as well.

*Fig.4:*
-   Please provide a useful web page or reference for "Stamen Design" (package/tool) in the legend and therefore in the reference-list.
-   Please provide the reference for the vectorial shapefile / grid file of Schmid 2004 faults geometry.
-   The authors should also display the classic longitude and latitude section profiles displaying the final seismicity at depths.
-   Please add the station marker (empty triangle) to the legend.

*Fig.5:*
-   The y scale for the "Day of the Weeks" panels (left sides) and "Hour of day" panels (right-side) should be equal for a clearer comparison.
-   Please add an example of anthropogenic event detection and filtering.

*Fig.6:*
-   Remove the bracket around relative.
-   Would be nice to have a and *b* value listed separately and not in the equation-style in the legend.

*Fig.8:* I like the color differentiation (in map) for the projected earthquakes in each panel!
-   Please remove the fault polarity from the straight-slip faults. The authors should either plot the geological dynamic *on each* fault front (that would also be a nice addition to help the discussion) or remove this information completely. This comment is valid for Fig.4 as well.
-   A magnitude scale legend is missing on the map, please add it.

*Fig.A4:* Y axis label(s) missing.

*Fig.A9:* The map should have lon/lat measurements and a magnitude scale legend.

**Additional comments:**
-   As a general comment, is there even an Appendix B? Otherwise, one could remove the Appendix nomenclature and leave the standard "Supplementary Information" one. In any case, I would change the naming of the appendix figures from Figure A1, A2 … to Figure F1, F2 etc. to avoid misunderstanding with the text blocks also named A*.
-   P2-L51 GPU acronym should be fully extended 2 lines before.

- P2-L54: … The extended set of P and S arrival times **are** then …
- P10-L23: Lower magnitude of completeness (not higher)
- In the reference list, Käestle et al. (2021) is misplaced (P30-L473). Should be placed after Kästle et al. (2020).
- The authors should think of merging section 2 and 3 in a more suitable "Data & Methods". Section 2 is too short in my opinion to stand alone, plus it is not really detached in terms of contents from section 3.

---

## Author Comment (AC1)

**Referee Comment #1**

In this study, the authors take advantage of a temporary, dense and relatively large seismic array (Swath-D Network) to build a new earthquake catalog and improve the current knowledge of the seismicity in the eastern and southern Alps. The authors present the results of a thorough analysis mixing automated and manual methods with which they detected and located 4 times more events than existing catalogs. The manuscript is well organized and written. I enjoyed reading about sophisticated methodology that doesn't involve machine learning. I only have minor comments about the methodology and some surprising results.

**Comments**

- Did the INGV and SED agencies use the same data set when building their catalogs? I find quite surprising that after the energy-based detection there are only 286 new events.

  *Yes, the seismic networks that are operated by the local agencies are publicly available. The agencies therefore have access to the same data and also state on their websites that they use data from external networks. Combining the local networks, the agencies have quite impressive detection capabilities in the area. This explains the relatively low number of additional energy-based detections. Note that this is only one of the first steps in our workflow, and the majority of events is detected by template matching.*

- Lines 104 and 296, you mention the correlation coefficient but don't give details about how many stations are involved in its computation. I think this information is important to understand what the value means.

  *The cross-correlation coefficient referred (line 104, 296) refers to the median of the three largest three values, using a maximum of 15 stations closest to the event, depending on availability. This has been clarified in the manuscript at the locations mentioned above, as well as in the figure captions of Figures A5 and A7.*

- Line 140 "Additionally, the number of I/O operations is reduced to a minimum by loading each continuous data trace only once." I don't really understand the meaning of this statement because I don't see why one would load the data several times?

  *Because our machine, especially the GPU, has limited memory, only a small number of templates and a small amount of continuous data can be loaded at once. In our code, the set of continuous data traces are held in memory while the templates are cycled (reloaded for each new set of continuous data traces). This is cheaper than cycling through the continuous data, but has some limitations. For example, if one were to use a summed CC approach, it would be necessary to load the data template-wise (i.e. load traces from different stations simultaneously), or to write the continuous cross-correlation functions to disk (which would take up a huge amount of space). Both of these options would require much more reading and writing.*

- The detection threshold is set in a very arbitrary way: How much is 0.5 in terms of standard deviations? CC detection thresholds are usually defined upon the root mean square or median absolute deviation of the CCs so that they correspond to a given p-value of false detections assuming a gaussian distribution. Based on my personal experience, 0.5 can also be very conservative. What is the motivation for not summing the CCs and applying the threshold on the summed CCs? Array techniques are all based on the idea that signal-to-noise ratio of a sum increases as N where N is the number of traces. But by applying the detection threshold on single stations, one loses the full benefits of network information.

  *We have experimented with using Median Absolute Deviation (MAD), as well as a summed CC approach. Both of these options reduced the stability of our results. In general, our CC threshold of 0.5 corresponds to an MAD of about 7-9. However, the CC obviously has a maximum of 1.0 when the waveforms are exactly the same, whereas the maximum of the MAD varies greatly over time and from station to station, making it difficult to set a universal threshold. Our first attempts at applying template matching to our dataset revealed that an enormous amount of triggers were due to station noise. The best way to stabilise the results was to use three independent station measurements instead of a combined network value. This greatly reduced the number of false triggers. It also has the advantage of being much faster, because we don't need to store all the continuous cross-correlation functions (see answer to previous question).*

- Lines 166-167: For phase picking, the authors use higher frequencies for S waves than for P waves. What motivated this decision? It is unexpected since P waves carry higher frequency energy (the P-wave corner frequency is about 50% higher than that of the S wave).

  *This was a typing error. We use the frequency band from 1-20 Hz, whereas for S-waves, frequencies between 1-12 Hz are used. Thanks for pointing this out.*
- Lines 178: I got a bit lost in these explanations. We start reading about picking on the template events, then on the newly detected events, and in this paragraph the authors go back to hand picks, which I thought were only for template events? I found this paragraph hard to understand so perhaps it is worth editing it.

  *Thanks for this suggestion. The paragraph was rewritten to make it clearer, and the previous two paragraphs were also revised slightly to improve the structure of the section. As for your question: only the master events are hand-picked. This is the first step. In the second step, these handpicks are used to find picks for the detected events by cross-correlation. However, changing station availability might therefore limit the number of picks for detected events. We therefore apply a third step, where a STA/LTA trigger is applied to traces that might be available for the detected event only (and not for the master event).*
- Magnitude estimation. Figure A5 shows some surprising observations. The lack of knowledge of what the CC means exactly (see comment above) might partly feed this comment. For event pairs with CC > 0.9, even within the reduced frequency band of 2-8 Hz, I don't see how the magnitude difference can be up to 2 units. Could it be caused by errors in the magnitude computation?

  *An explanation of the definition of the CC we used has been included here and in the figure caption (Fig. A5). We do not attribute the variation in magnitude to errors in the calculation. The frequency band used for cross-correlation (2-8 Hz) is well below the corner frequency of the events in this magnitude range (~ 20 Hz for $M_W$ 2). The cross-correlations are therefore performed on the flat part of the frequency spectrum. On the recommendation of the second referee, we have changed the method for magnitude calculation to a standard local magnitude approach. This actually increases the magnitude difference slightly, because the magnitude now scales directly to the log amplitude. An example of this is shown in the figure below, where the left panel shows a trace from a master event with Ml 1.72, and the right panel shows a detected event with Ml -1.01. The two traces have a maximum cross-correlation coefficient of 0.91 although the maximum amplitude differs by a factor of 500, confirming the magnitude difference of log(500) = 2.7. This particular event pair has four stations with a maximum cross-correlation coefficient greater than 0.9, all of which show a similar difference in maximum amplitude.*

[Figure]

- Line 291: A mean location uncertainty of only 300m? Most earthquakes are located outside the network (Figure 4) so I would expect a much higher number.

  *Although it is true that some events are located slightly outside of the network, this is only a small fraction of the event catalogue. The density of the events within the network area is much higher, something that can not be directly derived from Figure 4, as there are very many overlapping points in the central parts of the map. On request of the second referee, we included a figure with the original NonLinLoc errors to support this claim.*

- Line 297: "upper bound" on location uncertainty. I would avoid making a quantitative statement from a hand wavy argument. Is the idea that events with CC>0.9 are more-or-less exactly co-located? In this case, shouldn't it be a lower bound (true uncertainty = within-family dispersion + NLLoc uncertainties)?

    *If we assume that events with CC>=0.9 are exactly colocated, we can attribute the dispersion to location uncertainty entirely. In reality, part of the dispersion may be natural (e.g. events are not exactly colocated), and in this case the contribution of location uncertainty would be smaller. Therefore we call it the upper bound.*

- It would have been nice to have the catalog as a supplementary file. The authors do say that they are willing to share the file upon request, but this seems incompatible with anonymity.

    *Yes, the final manuscript will contain a link to the event catalog. The statement referred to was intended for the referees during the review process.*

---

## Author Comment (AC2)

**Referee Comment #2**

The paper from Jan Hofman et al. focuses on the development of a new semi- automatic methodology for the creation of a consistent seismic catalog in the Eastern Alps, as part of the Swath-D project. The method involves both manual and automatic procedures, as well as the GPU usage to fasten up the process. This application aimed at precise arrival pick detections for P and S phases, thus producing improved absolute seismic locations. I found the paper well written in English, with room for improvements mainly for the methodological sections. I also appreciate the authors shifting most of the methods in the Appendices for a clearer view of the main goals. Eventually, I recommend the publication of this work after revisions.

**Main General Comments:**
- To meet the FAIR publications criteria, the authors should provide the final catalog with at all the needed metadata to replicate their results. The codes used in the study should be also distributed in one or more open access repository (i.e. GitHub, GitLab, Zenodo) at authors discretion. I think that especially for methodological paper, this latter point is a must.

  *The revised manuscript includes a link to the data repository DOI where the final catalogue can be downloaded. The publication was not intended to be a methodological paper. Therefore, most of the methodological figures appear in the appendix. The codes were not originally intended to be published, as they were developed and optimised to run on our particular hardware setup. Performance on other machines is not supported or tested. Documentation is also very limited. However, on the recommendation of the referee, we agree to publish the code in the same repository as the catalogue. A short readme will be included to help users understand the code.*

- In my opinion, the magnitude calculation and catalog completeness definition are a bit incomplete. First, the author should provide a clear value for the completeness consistently throughout the manuscript: the symbol ~ should not be used (i.e. P1-L3 / P9-L222/ P16-L311). Second, how is the completeness calculated? With which method? Is one or more statistical approaches involved? Finally -and most important- why not pursue a much more consistent strategy of reassessing the event local magnitude based on S-wave amplitude? Having already extracted the amplitude windows during the cross-correlation and repicking stages, it should be straightforward to apply a standard ML attenuation function and the catalog would highly benefit from it in terms of consistency and robustness, not to mention for the discussions section. One could then plot the master events initial magnitude against the recalculated one to validate a linear fit trend. I would recommend these calculations and to better validate the catalog completeness, although I leave the decision to the authors.

  *The motivation for our initial approach was to provide magnitudes that were consistent with those published by the agencies. However, we agree that providing local magnitudes is the more transparent and data-driven approach. It is also true that the agencies are likely using different scales and methods, as the magnitudes are not always consistent between the different catalogues. We therefore agreed to calculate local magnitudes based on the mean of the horizontal absolute S-wave maximum amplitudes. We applied a standard attenuation function, using the geometrical spreading and inelastic attenuation parameters derived for the area by Bragato and Tento (2015). Station correction terms were calculated to remove the residual mean magnitude for a subset of good quality events (master events). An additional estimate of the moment magnitude is shown in Fig. A6 in the appendix for comparison, as the local magnitude scale tends to underestimate the seismic moment for low-magnitude events (Mw < 4) (Mufano et al., 2016). The magnitude of completeness in both magnitude scales is estimated from the maximum curvature of the frequency magnitude distribution. This is now mentioned in the text and the '~' symbol has been removed. All figures showing magnitudes have been updated to show the new local magnitudes, and all references to magnitudes in the main text or figure captions have also been updated.*

  *Bragato, P. L. and Tento, A.: Local magnitude in northeastern Italy, Bulletin of the Seismological Society of America, 95, 579–591, 2005.*

*Munafò, I., Malagnini, L., and Chiaraluce, L.: On the Relationship between Mw and ML for Small Earthquakes, Bulletin of the Seismological Society of America, 106, https://doi.org/10.1785/0120160130, 2016.*

- Still talking about magnitudes...citing appendix A4: "Event magnitudes for detected events are computed relative to the magnitudes in the publicly available event catalogues, based on peak S-wave amplitude ratios. For each station, peak S-wave amplitudes are determined for all events with known magnitudes recorded by the station". In the follow up equation, though, I see the least-square fit based on master event station magnitudes. I think this assumption could lead to biases. Indeed, the master events extracted from national bulletins are obtained with different local magnitude scales parametrization. Please provide some reasoning for your decision and try to expand the discussion in the Appendix A4.

    *This point is no longer relevant since we have now calculated local magnitudes that are completely independent of magnitudes published in the national bulletins.*

- I would try stress a bit more the discussion over the improvement with the authors GPU support solution. Did the authors tried to benchmark their approach with other methods? Or even compare it with the serialization or standard parallelization approach? Some absolute and quantitative number would be nice to have for the reader.

    *Tests were made with existing template matching codes, but they either did not provide satisfactory results or were too slow to process our dataset in a reasonable amount of time. We therefore decided to develop our own code. We did not benchmark the code. However, to give some reference numbers: in a very primitive, serial template matching approach, a speed of about one cross-correlation per second can be achieved. By one cross-correlation we mean the cross-correlation of a single (one station, one channel) 10 second template trace with a 24 hour continuous data trace. For our dataset (28k template traces, ~ 2y continuous data) this extrapolates to about 236 days. Using our GPU approach, we were able to achieve speeds of about 100 cross-correlations per second. We have included a statement about the relative speedup in the 'Template Matching' section.*

- Another important yet underexplained part of the methodology is the anthropogenic / natural seismic event definition (and therefore filtering). P2-L56 "We identify these anthropogenic signals based on their typical temporal signatures and confirm their origins using satellite images.". Is it really based only on temporal signatures? How where the satellite images compared? The authors should provide an example of anthropogenic event detection and filtering to complement Figure 5. This stage is as important as a good picking refinement, and I think it should deserve more space in the manuscript. Did the authors adopted any criteria to stay on a more conservative scenario (i.e., losing more seismic events compared to risking having anthropogenic noise labeled as seismic events)?

    *We agree that an example might clarify the explanation here. We have tried extensively (in the frame of a Bachelor's thesis in our project) to detect non-seismic signals such as quarry blasts and explosions from the event waveforms by extracting many different waveform features and performing a principle component analysis on a labelled test dataset. Unfortunately, this did not produce a satisfactory result. Even by manual inspection of the waveforms, the event classes are very hard to distinguish in many cases, because the signal-to-noise-ratio is usually low. We therefore decided to look at the temporal signatures. In particular, the day of week histogram and the hour of day histograms turned out to be very useful. Quarry explosions are usually set off at the same time of day (e.g. 15:00) on weekdays only, and are therefore quite easy to identify. In case of doubt, satellite imagery can confirm the presence of quarries or construction sites at the location of the events. As requested, a figure showing an example of a non-seismic event family has been included in the appendix (Figure A5).*

- I think the author could provide an additional figure (in the supplementary) showing the x-y-z-ot uncertainty distributions of the final catalog events, to validate the average errors declaration.

    *A figure showing the dimensions of the 68% confidence ellipsoids was added in the appendix (Figure A9).*

**Additional comments:**
- Fig.1:

- A differentiation with marker type (i.e., square or circles) is helpful to discern between the permanent and temporary stations. Please group them accordingly.

  *Thank you for the suggestion, the marker type for permanent stations was changed to a square to help distinguish the permanent and temporary stations (Figure 1).*

- Please provide a useful web page or reference for "Stamen Design" (package/tool) in the legend and therefore in the reference-list.

  *A link to the webpage is now included in the captions of all figures using this map background. (Fig. 1, Fig. 4, Fig. 8, Fig. A5, Fig. A12)*

- The number 198 (P2-L48) correctly sum up the stations number listed in the legend. The authors, though, also mention that the SWATH-D is composed by 151 stations (P2-L32). In the figure's legend is written 147. Please double check the numbers and correct accordingly both in captions and in text.

  *This is correct. Four stations (D030, D142-D144) were deployed up to 10 months later than the rest of the network. The same is true for the eastward extension of the network with stations D154-D163. We chose not to use these to have a more stable network geometry over time. We extended the explanation on this issue in the Data section. Thank you for pointing out this apparent inconsistency.*

- The Z3 and ZS temporary network listing and their respective citations are missing in the figure's caption. Please add them.

  *The references for Z3 and ZS are now included in the figure's caption (Figure 1).*

- In my opinion, displaying the fault systems is a bit off the figure's scope, as the main aim is to describe the station network. The authors correctly display and describe the fault system in Fig.4 and Fig.8, and therefore I would suggest the authors to remove them from this figure.

  *We agree that displaying the fault systems in this figure is not necessary, and maybe even distracting. We removed them from the figure (Figure 1).*

- I think it would be nice for the reader and for the completeness of the manuscript to provide a figure in the supplementary material with the initial seismicity geometry (the one from the agencies bulletin) in different colors. Possibly, sided with the merged seismicity. I strongly recommend such a figure.

  *We think that such a figure would tempt the reader to compare the seismicity catalogues from the different national bulletins, and this is not the point of our paper. In fact, the spatial resolution of the (combined) catalogues is very good, and the gain from our study is mostly in the number of events and the completeness. Both of these features are not very well represented by a map view comparison. If the reader is interested, all agencies provide interactive mapping tools on their websites.*

- Fig.2: Is there a reasoning why the authors use different colors (blue / gray) and occasionally grouping in gray box? In the manuscript, the method part is divided in 3 main blocks, I would recommend the authors to follow that flow consistently in the figure as well.

  *The blue boxes indicate intermediate results after applying a processing step. The 'public catalogues' box is grey since it is not a result of our work. The grouping means that the method (event clustering) was applied to both items. A '+' is now added between the grouped items to make this point clearer. As for the three subsections in the Methods section, they are represented by the three columns in this figure. A dotted, grey outline is now added to each column, with a reference to the corresponding methods subsection at the bottom of each outline. Hopefully, this helps to understand the logic of the figure (Figure 2).*

- Fig.4:
  - Please provide a useful web page or reference for "Stamen Design" (package/tool) in the legend and therefore in the reference-list.

    *A link to the webpage is now included in the captions of all figures using this map background. (Fig. 1, Fig. 4, Fig. 8, Fig. A5, Fig. A12)*

  - Please provide the reference for the vectorial shapefile / grid file of Schmid 2004 faults geometry.

    *A link to the data repository where the shapefile can be downloaded is now included in the figure caption (Figure 4).*

  - The authors should also display the classic longitude and latitude section profiles displaying the final seismicity at depths.

*Because the study area is rather large and heterogeneous, longitude and latitude versus depth profiles for the entire region do not really facilitate any interpretation. We therefore chose to include only the depth histogram in this figure. However, in the data repository, we will upload two videos with sliding north-south and east-west cross-sections across the entire map in 0.01 degree increments. Additionally, Figure 8 provides a good view into the depth of seismicity in different parts of the region.*

- Please add the station marker (empty triangle) to the legend.
  *A legend entry for the station marker was added (Figure 4).*
- Fig.5:
  - The y scale for the "Day of the Weeks" panels (left sides) and "Hour of day" panels (right-side) should be equal for a clearer comparison.
    *The point of this figure is to show the temporal patterns in the event origin times and not to compare the absolute numbers. The dataset contains less anthropogenic events than earthquakes, so the bars in the lower panels would become much smaller. Also, there are obviously less events in one hour than in one day. This is just because of the smaller bin size. Keeping the y-scale constant in all of the panels would make it harder to see the patterns that we are aiming to emphasise.*
  - Please add an example of anthropogenic event detection and filtering.
    *This is the same point as the fifth points in the Main General Comments part. A figure with an example was added in the appendix (Figure A5)*
- Fig.6:
  - Remove the bracket around relative.
    *The word relative was removed entirely (Figure 6).*
  - Would be nice to have a and b value listed separately and not in the equation-style in the legend.
    *The b-value was mentioned in the main text already, but now both the a- and b-value are mentioned in the figure caption as well.*
- Fig.8: I like the color differentiation (in map) for the projected earthquakes in each panel!
  - Please remove the fault polarity from the straight-slip faults. The authors should either plot the geological dynamic on each fault front (that would also be a nice addition to help the discussion) or remove this information completely. This comment is valid for Fig.4 as well.
    *Thank you for the positive feedback. The fault traces were added to provide a basic geological reference frame. They mark the boundaries between the different tectonic units, and a clear correlation with the seismic activity can be seen this way. We think that this is very helpful for the interpretation of the results. The fault polarities may be helpful for readers that are interested, and we think that for readers that are not interested, they are not distracting.*
  - A magnitude scale legend is missing on the map, please add it.
    *A magnitude scale legend is now included on the map. Additionally, topography profiles were added to each of the cross-sections (Figure 8).*
- Fig.A4: Y axis label(s) missing.
  *The y-axes in this figure represent normalised amplitude. Labels are now added (Figure A4).*
- Fig.A9: The map should have lon/lat measurements and a magnitude scale legend.
  *This figure was updated and now includes a latitude/longitude grid, as well as a legend with symbols to indicate the magnitude scale (Figure A12).*

**Additional comments:**
- As a general comment, is there even an Appendix B? Otherwise, one could remove the Appendix nomenclature and leave the standard "Supplementary Information" one. In any case, I would change the naming of the appendix figures from Figure A1, A2 ... to Figure F1, F2 etc. to avoid misunderstanding with the text blocks also named A*.
  *From the Solid Earth manuscript composition guidelines, we understand that supplementary material is published as separate documents, and intended for videos, very large images, or for example code. Appendices are intended for additional figures, or to provide extra detail and support for experts. The naming of the appendices and the figures and sections therein is also specified in the guidelines.*

- P2-L51 GPU acronym should be fully extended 2 lines before.
  *The paragraph was slightly edited so that the acronym is expanded where it is first mentioned.*
- P2-L54: ... The extended set of P and S arrival times **are** then …
  *We have adopted this suggestion.*
- P10-L23: Lower magnitude of completeness (not higher)
  *This sentence discusses the b-value of the frequency magnitude distribution, that is slightly higher for the template matching catalogue compared to the master events. The magnitude of completeness is in fact lower. The paragraph was rewritten to reference the numbers more clearly.*
- In the reference list, Käestle et al. (2021) is misplaced (P30-L473). Should be placed after Kästle et al. (2020).
  *Thank you for pointing this out. The reference is now placed after Kästle et al. (2020).*
- The authors should think of merging section 2 and 3 in a more suitable "Data & Methods". Section 2 is too short in my opinion to stand alone, plus it is not really detached in terms of contents from section 3.
  *We agree that the Data section his quite short, but the contents are clearly distinct from the Methods. We feel that merging these two sections does not improve the readability of the paper. The Data section helps guide a reader that is quickly scanning the paper to look for the data sources.*

---

## Author Response (AR2)

**Referee Comment #1**

The Authors have comprehensively responded to the previous reviews and have addressed all the issues satisfactorily. This has improved the manuscript to the expected level. There are a few minor and technical details that remain, which I suggest addressing before the manuscript publication.

**Comments**

- P3L68: "ten additional stations (D154-D063)". Probably the author meant "D154-D163"

  *Yes, this was a typing error. We meant "D154-D163" as suggested. Thanks for pointing this out.*

- P12L244: "… based on the maximum curvature of the FMD." It misses the paper citation of the method (Wiemer, S. and M. Wyss (2000). "Minimum Magnitude of Completeness in Earth-quake Catalogs: Examples from Alaska, the Western United States, and Japan". BSSA). This comment is also valid for Figure 6 and A6 captions.

  *We now included a reference to Wiemer and Wyss (2000) at the three locations mentioned above.*

- P25L405: "amplitudes on the vertical channels …" should be corrected in "… amplitudes on the horizontal channels …"

  *Thanks for pointing out this error. We changed "vertical" to "horizontal" in this sentence.*

- I suggest the authors homogenize the X-axis plot limits of Figure 6 and A6 to have a straightforward comparison between magnitude scales. I finally recommend authors include both ML and Mw in the distributed catalog file.

  *Figure A6 now has the same X-axis limits as Figure 6 to make the comparison between both figures easier. The final catalogue in the data repository contains both Ml and Mw.*

- The DOI provided https://doi.org/10.5880/fidgeo.2023.024 is currently not working (last try on August 25th, 2023). Please provide a functioning DOI link for the repository (catalog and codes) upon publication.

  *Our data repository manager recommended we wait for the 'accepted' status before publishing the dataset, because it is not possible to make any changes after publication. We are sorry for the confusion. As soon as the manuscript is accepted for publication, we will activate the link to the dataset.*